# Anomaly based multi-stage attack detection method

**Wei Ma**[1,3]**, Yunyun Hou**[1]*****, Mingyu Jin**[2]**, Pengpeng Jian**[1]

**1** North China University of Water Resources and Electric Power, Zhengzhou, Henan, China, **2** Zhengzhou University of Light Industry, Zhengzhou, Henan, China, **3** Zhengzhou Normal University, Zhengzhou, Henan, China

* x20211090823@stu.ncwu.edu.cn

**Data Availability Statement:** The data that underlie the results presented in the study are stored in a public repository and are accessible through the following links: 'https://archive.ll.mit.edu/ideval/data/2000data.html,' 'https://www.unb.ca/cic/

## Abstract

Multi-stage attacks are one of the most critical security threats in the current cyberspace. To accurately identify multi-stage attacks, this paper proposes an anomaly-based multi-stage attack detection method. It constructs a Multi-Stage Profile (MSP) by modeling the stable system's normal state to detect attack behaviors. Initially, the method employs Doc2Vec to vectorize alert messages generated by the intrusion detection systems (IDS), extracting profound inter-message correlations. Subsequently, Hidden Markov Models (HMM) are employed to model the normal system state, constructing an MSP, with relevant HMM parameters dynamically acquired via clustering algorithms. Finally, the detection of attacks is achieved by determining the anomaly threshold through the generation probability (GP). To evaluate the performance of the proposed method, experiments were conducted using three public datasets and compared with three advanced multi-stage attack detection methods. The experimental results demonstrate that our method achieves an accuracy of over 99% and precision of 100% in multi-stage attack detection. This confirms the effectiveness of our method in adapting to different attack scenarios and ultimately completing attack detection.

## Section 1: Introduction

In cyberspace, network attacks have consistently been one of the most pressing security concerns. Intrusion Detection Systems (IDS) can mitigate this issue to a certain extent, but multi-stage attacks (MSA) [1] remain highly destructive. MSA is a complex attack paradigm consisting of multiple individual attack steps. Unlike single-step attacks, MSA often has well-defined attack objectives (mostly high-value targets, such as core corporate assets) and results in substantial damage (as seen in the 2017 WannaCry incident). Advanced persistent threat (APT) attacks are one of the prominent attack methods in this category [2].

IDS can provide alert information for individual steps within MSA, and this alert information can be used for aggregation, inference, and prediction to counter the threats posed by MSA. Analyzing MSA using IDS alert information has been a hot topic in recent years. Some researchers have employed log correlation analysis methods to recreate MSA attack scenarios.

datasets/ids.html,' and 'https://www.unb.ca/cic/datasets/ids-2017.html'.

**Funding:** This work is supported by the National Natural Science Foundation of China (No. 6210701),the funding amount is 300,000. Henan Key Research Projects of Universities (No. 23B520010),the funding amount is 100,000, and Key R&D and Promotion Projects of Henan Province (No. 212102210100), the funding amount is 50,000. The funders had no role in study design, data collection and analysis, decision to publish, or preparation of the manuscript.

**Competing interests:** The authors have declared that no competing interests exist.

The key lies in analyzing the cause-and-effect relationships between alert information before and after, thus establishing the correlation between alert information [3, 4]. This approach aids in reconstructing the complete attack path and attack scenario of MSA. Other researchers have used machine learning and artificial intelligence methods to mine the knowledge embedded in alert information and utilize this knowledge to detect, infer, and predict MSA. Algorithms employed include Hidden Markov Model (HMM), Support Vector Machine, Decision Trees, Bayesian Networks, and Deep Neural Networks, among others [5–10]. However, both of these approaches have their limitations. Firstly, they depend on prior knowledge, i.e., knowledge of known attack paradigms. For log correlation-based methods, prior knowledge is required to extract and analyze the relationships between alert messages, and for learning-based methods, prior knowledge is necessary to build models. Secondly, the information contained in alerts is often protocol-specific, discrete, and low-level. These two shortcomings significantly impact the detection capabilities of the models.

An example can better illustrate the aforementioned limitations. In the 2018 attack on the global SWIFT system initiated by APT38 [11], attackers initially launched spear-phishing or watering hole attacks to target specific objectives, completing the initial penetration. They then moved laterally within the penetrated network, seeking opportunities to gain SWIFT system terminal privileges, ultimately altering SWIFT transaction information to execute the attack, as illustrated in Fig 1.

In this three-stage APT attack example, attackers do not strictly adhere to this sequence of attack actions. For instance, attackers may, after laterally moving within the target network (see ③ in Fig 1) from the first victim (see ② in Fig 1), return to the first step to identify new victims (see ① in Fig 1). Such deviations from known attack processes can introduce disruptions to prior knowledge, thereby affecting the accuracy of correlation analysis and attack detection.

To overcome the mentioned limitations, and drawing inspiration from [12–14], this paper introduces an anomaly-based multi-stage attack detection method. In contrast to attack behaviors, the system's normal behavior is inherently more stable and reliable. Therefore, leveraging the concept of "staging," we model the system's normal behavior in stages without the need for prior knowledge. Firstly, we process alert texts using the Doc2Vec method, transforming them into alert vectors. Subsequently, conducting cluster analysis on alert vectors to segment the system's normal state into several stages. Following that, utilizing HMM algorithms to construct the system's Multi-Stage Profile (MSP) and utilizing MSP to determine whether the system is under attack. Compared to previous methods, our approach can extract deep-seated

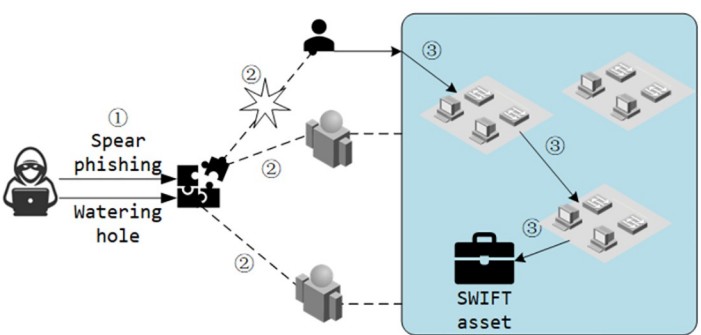

**Fig 1. An example of an APT attack.**

information from alerts, analyze the relationships between alert entries, and accomplish model building without the need for prior knowledge using unsupervised clustering methods.

The main contributions of this paper are as follows:

- An original concept of Multi-Stage Profile (MSP) was introduced. Borrowing from the idea of dividing attacks into multiple stages, the paper performs staged modeling of a system's normal state. This enhances the model's adaptability and generalization to different network environments and attack types, improving its overall adaptability and generalization.

- To accommodate various attack scenarios, a novel method for adaptive acquisition stages is proposed. This method utilizes Doc2Vec to deeply extract information from alert messages, preserving the interrelations between alerts more effectively. Subsequently, it clusters vectorized alert information, grouping semantically similar alerts into the same stage. This method demonstrates adaptability to various attack scenarios.

- An original concept of Generating Probability (GP) is introduced, used to calculate the probability of an alert sequence occurring. We employ GP to establish anomaly thresholds, enabling attack detection.

- Comparative experiments conducted on three different datasets (DARPA2000, ISC-XIDS2012, CIC-IDS2017): Extensive experimental results illustrate that this method achieves an accuracy of over 99% in detecting multi-stage attacks across various datasets, outperforming the latest methods.

The remaining structure of this paper is organized as follows: Section 2 presents a literature review related to multi-stage attack detection. In Section 3, we detail the design process of the proposed anomaly-based multi-stage attack detection method. Section 4 includes an evaluation of the model, including a comparison with the latest multi-stage attack detection methods. Finally, Section 5 offers a summary of the paper.

## Section 2: Related works

In recent years, various methods have been proposed to address issues related to multi-stage attack detection, with Bayesian models and HMM being the primary approaches for detecting multi-stage attacks. Ren et al. [15] introduced a multi-stage attack detection method based on Bayesian models, which divides the detection process into two stages. Firstly, it employs Bayesian networks to automatically extract correlations and constraints between alerts, testing different features to find the most accurate descriptors for attack stages. Then, based on the selected features, it extracts attack scenarios from the alert stream. Marchetti et al. [16] proposed the use of Bayesian models to calculate alert correlations, identify whether alerts belong to the same attack scenario, and generate an alert correlation graph. However, these methods reconstruct attack scenarios through alert correlations, requiring a significant amount of prior knowledge and increasing the complexity of maintaining a secure system. Moreover, these methods merely replicate attack scenarios and do not detect the stages at which attacks occur.

As early as 2003, HMM was employed to address the issue of MSA detection. HMM is a dual stochastic process [17], and in the field of statistical machine learning, it is considered one of the most suitable techniques for multi-stage attacks detection. The main reason for this is its mathematically tractable form for analyzing input-output relationships and generating transition probability matrices based on training datasets. D. Ourston et al. [18] utilized HMM for detecting multi-stage attacks and compared it with two other classical machine learning algorithms, decision trees, and neural networks. The results showed that HMM outperforms decision trees and significantly surpasses neural networks in multi-stage attacks detection.

Chen et al. [19] proposed the introduction of HMM in the cloud for attack sequence detection, defining the detection of multi-stage attacks as a state-based classification model. Holgado et al. [20] provided a more detailed introduction to how HMM can be applied to multi-stage attacks. They defined states based on Common Vulnerabilities and Exposures (CVE) statistics, combining multi-stage attack data with the CVE database. They explained how to construct HMM using supervised and unsupervised learning methods, namely using the Baum-Welch algorithm or statistical frequency methods to train model parameters. The Viterbi algorithm and forward-backward algorithm can be used to determine the most likely attack stages. Suratkar et al. [21] introduced an HMM-based Host Intrusion Detection System (HIDS) model that consists of an anomaly detection module built using Long Short-Term Memory (LSTM) and multiple HMM modules for multi-stage attack detection. Due to the significant impact of HMM parameters on detection performance, Chadza et al. [22] designed an effective detection framework combining transfer learning and HMM. They trained HMM on labeled data and transferred the learned parameters to new tasks. Unlike other discrete modeling techniques, HMM excel in hidden states and transitions, thereby eliminating the need for complete information before attack detection.

Furthermore, With the advancement of big data technology [23], some deep learning methods have been applied to multi-stage attack detection. Deep learning approaches can overcome some limitations of traditional shallow machine learning, capturing deep-seated features within the data [24], and enhancing detection performance [10]. Vinayakumar et al. [25] introduced a deep learning framework for detecting zombie networks, which operates at the application layer of DNS services. This framework works by distinguishing between normal behavior and zombie network behavior. Sudheera et al. [26] conducted research on multi-stage attacks and proposed a distributed multi-stage attack detection method. Their work addressed the spatiotemporal challenges of zombie network attacks. They used alert-level and pattern-level information as features and employed machine learning methods to identify various attack stages within generated alerts. Xu et al. [27] designed an LSTM network based on multiple feature layers. They introduced a stage feature layer to store and compute historical data to identify different stages of multi-stage attacks with varying durations. Then, they used a time series feature layer to link independent attack stages and analyze whether the current data is within a certain attack cycle. However, these methods struggle to detect unknown attack paradigms and have lower detection effectiveness for new attack behaviors.

In the method proposed in this paper, we leverage the relative stability of the system's normal state to construct an MSP of the normal state. This MSP allows us to label alerts that do not conform to the normal state as attacks, thereby achieving multi-stage attack detection. The method introduced in this paper can directly build statistical detection models from the raw data of alerts without the need for additional expert knowledge or specific attributes. Additionally, it can detect the specific stages at which alerts occur, while also addressing the limitation of traditional methods in detecting unknown attack paradigms.

## Section 3: Anomaly based multi-stage attack detection method

### 3.1 Method model overview

The overall architecture of the method proposed in this paper, as shown in Fig 2, consists of four steps: network data acquisition (phase 1), alert preprocessing (phase 2), establishment of MSP (phase 3), and attack detection (phase 4).

Firstly, in phase 1, First, in Phase 1, we designed an automated data acquisition method to obtain alert information from network traffic. Deployed IDS in the network continuously analyze traffic data captured from the network environment and generate alerts when suspicious

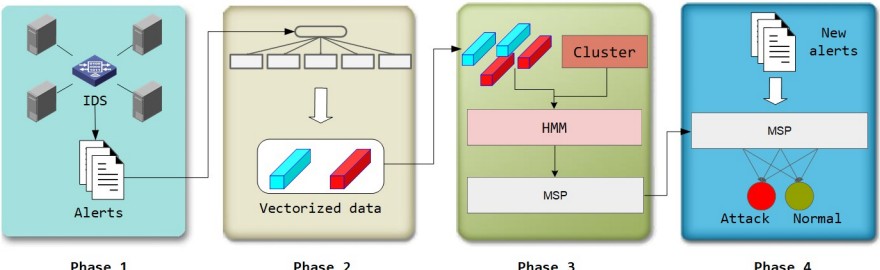

**Fig 2. Anomaly-based multi-stage attack detection framework.**

packets are detected based on predefined rules. IDS may not detect complete multi-stage attacks, but when attackers attempt to infiltrate through multiple attack stages, IDS may capture individual attack actions and issue corresponding alerts. In Phase 1, the system primarily faced performance pressure stemming from IDS traffic analysis, and therefore, we adopted an offline analysis strategy to avoid impacting the overall system performance.

However, it is worth noting that IDS systems generate a significant number of false positives. These alerts result from the inability of the alert generation rules to distinctly differentiate between normal and malicious activities within the network, and thus, do not represent genuine security threats [28]. Nevertheless, these non-attack stage alerts often carry information about the system's activity patterns, serving as a means to describe the system's normal state. Alerts generated by IDS are stored in the Alerts Database, which encompasses alert data from the system's normal state (referred to as non-attack stage alerts) and alert data from multi-stage attack states (referred to as attack stage alerts in this paper).

In phase 2, we introduce a method for alert preprocessing with the goal of transforming the text-style alert data generated in Phase 1 into data that can be used by machine learning algorithms. Since HMM cannot directly process alert information, we convert the alert data from the Alerts Database into vectors using the Doc2Vec algorithm. This allows us to extract deep-seated information from the alerts and analyze the associations between alert entries for further processing. The performance overhead in Phase 2 primarily stems from the training of the Doc2Vec model. Therefore, similar to Phase 1, we adopt an offline training strategy in Phase 2.

Then, the proposed MSP is constructed in Phase 3. We initially use a clustering approach to automatically obtain the stage division of normal alert vectors, which is then mapped to hidden states in the HMM to complete the construction of MSP based on HMM. Training the clustering model and HMM introduces a significant performance overhead in this phase. Therefore, similar to previous phases, we also employ an offline training approach.

Finally, in Phase 4, we perform online detection of alert data using the constructed MSP. The probability generated by the MSP is used as the basis for determining anomalies. This probability is compared to a predefined threshold to decide whether the sequence is anomalous. If it is, then the alert is marked as an attack stage alert. In contrast to Phases 1, 2, and 3, the detection process in Phase 4 is conducted online, using the MSP model obtained in the offline training of Phase 3. Online detection is nearly real-time, and it imposes relatively low performance overhead.

To provide a better understanding, we summarize the formula symbols and their descriptions used in various stages of the model in Table 1.

**Table 1. Symbol table.**

| Symbol | Description |
|---|---|
| $\{a_1, a_2, \ldots, a_n\}$ | Alarm sequence, where $a_i$ indicates the $i$-th alert in the sequence |
| $w_1, w_2, \ldots, w_T$ | The training term obtained from the alert sequence, $T$ represents the number of training words |
| $argmax$ | Mean logarithmic probability |
| $\log p(w_t|w_{t-z}, \ldots, w_{t+z})$ | Represents the probability of output $w_{t-z}, \ldots, w_{t+z}$ given the input sequence $w_t$ |
| $W$ | Word vector |
| $D$ | Paragraph vector, which represents the tag vector for each alert |
| $h = contact(W, D)$ | It is constructed by word vector and paragraph vector |
| $U, b$ | $softmax$ parameters |
| $y_i$ | The non-normalized logarithmic probability of the output word |
| $S = \{s_1, s_2, \ldots, s_N\}$ | A set of states, representing a cluster containing $N$ different states, which are represented as attack phases or clusters (represented as clusters in this article) |
| $V = \{v_1, v_2, \ldots, v_M\}$ | The observation set contains $M$ different alerts |
| $\pi = (\pi_i = P(i_1 = s_i)), i = 1, 2, \ldots, N$ | Initial state probability vector |
| $A$ | state transition matrix |
| $B$ | Observation probability matrix |
| $\lambda = (A, B, \pi)$ | HMM model with parameters $A, B, \pi$ |
| $d_i$ | Represents the number of states $i$ in the sequence |
| $D_t$ | Represents the number of the entire sequence |
| $K$ | Number of clustering centers |
| $d_{ij}$ | The distance between the sample and the cluster center |
| $E$ | ERF error function |
| $S(i)$ | Average profile coefficient |
| $stage_i$ | Indicates whether the alarm of the $i$ sample is in the attack phase |

## 3.2 Alert preprocessing

The semantic description of alerts can be seen as a sequence of statements, and if the context of two alert descriptions is similar, it can be considered that they have similar semantics. In multi-stage attacks, the attacker's actions are intentional, and the alerts from attack stages also exhibit certain characteristics. Similar attack methods result in similar alert information. Therefore, by learning alert semantic representations from a large number of alert sequences, it is possible to effectively represent alerts.

In order to extract the semantic description of alerts and use it for further computation, we need to represent them in vectorized form. There are various methods for vectorizing semantic descriptions, such as the Bag-of-Words model [29], One-hot Encoding [30], and others. However, these methods do not capture the relationships between words in the alert information, and their sparse representation can lead to the curse of dimensionality. Word2Vec [31, 32] addresses the dimensionality issue but loses sequence information by averaging word vectors. When using Word2Vec to compute text similarity, keyword extraction algorithms may not perform accurately. To address these issues, the alert preprocessing model proposed in this paper utilizes the Doc2Vec model to transform alert descriptions into low-dimensional continuous values, mapping semantically similar alert descriptions to nearby positions in the vector space, thus extracting semantic knowledge from the alert descriptions.

The Doc2Vec model is capable of representing variable-length sentences, paragraphs, or documents as fixed-length vectors. This model not only utilizes the semantic information of

words but also incorporates information about the context and sequence of words [33]. The vectorization method used in this paper is based on the PV-DM (Distributed Memory Model of Paragraph Vector) model, which is derived from the skip-gram model by adding paragraph vectors. Specifically, given an alert sequence $[a_1, a_2, \ldots, a_n]$ and the training word $W_1, W_2, \ldots, W_T$, the objective of the word vector model is to maximize the average log probability, as described in Eq (1)

$$\text{argmax} = \frac{1}{T} \sum_{t=z}^{T-z} \log p(w_t | w_{t-z}, \ldots, w_{t+z}), \tag{1}$$

Where $T$ represents the number of training words, $z$ represents the window size, and $\log p(W_t | W_{t-z}, \ldots, W_{t+z})$ represents the probability of output $W_t$ given input sequence $W_{t-z}, \ldots, W_{t+z}$, where:

$$p(w_t | w_{t-z}, \ldots, w_{t+z}) = \frac{e^{y_{wt}}}{\sum_i e^{y_i}}, \tag{2}$$

Each $y_i$ is the unnormalized log probability of the output word, computed as:

$$y = b + Uh(w_{t-z}, \ldots, w_{t+z}; W, D), \tag{3}$$

Where $U$ and $b$ are softmax parameters, and $h$ is constructed jointly from word vectors $W$ and paragraph vectors $D$.

PV-DM has two key stages:

1. "Training Phase": Training word vectors, weights, and paragraph vectors based on observed alert information.

2. "Inference Phase": Adding additional columns to the paragraph vectors and obtaining paragraph vectors for new alert information while keeping the weights fixed using gradient descent.

Fig 3 illustrates the process of transforming alert information into vectors. Firstly, the alert information is tokenized, and word vectors $W$ (representing individual words) and paragraph vectors $D$ (representing each alert as a contextual vector) are trained. The network model parameters are updated using the cross-entropy loss function and gradient descent. Finally,

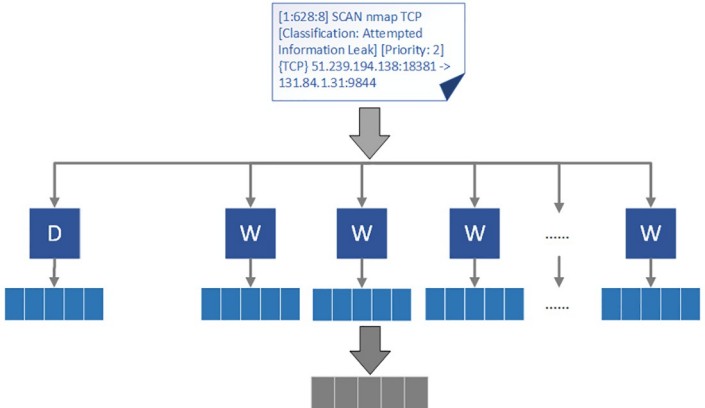

**Fig 3. Example of alert information transformed into vectors.**

the concatenation of word vectors and paragraph vectors serves as the embedding vector for each alert.

## 3.3 Multi-stage profile

**3.3.1 HMM.** HMM is widely applied in attack detection, and literature [18–22] has provided substantial evidence of the effectiveness of HMM in attack detection. HMM has unique advantages in stage modeling, as alert sequence data can be effectively matched with the HMM representation. Therefore, this paper constructs MSP based on HMM.

An HMM can be described by five elements: the set of possible states, the set of possible observations, the initial state probability vector, the state transition probability matrix, and the observation probability matrix. When applying HMM to multi-stage attack detection, each element is represented as follows:

1. The set of states, denoted as $S = \{s_1, s_2, \ldots, s_N\}$, consists of $N$ distinct states, and the state at the time $t$ is defined as $i_t \in S$. In the context of multi-stage attack detection, the states represent attack stages or clusters.

2. The set of observations, denoted as $V = \{v_1, v_2, \ldots, v_M\}$, consists of $M$ distinct alerts, and the observation at the time $t$ is defined as $o_t \in V$. In the context of multi-stage attack detection, the alerts generated by IDS are defined as observations.

3. Initial state probability vector $\pi = (\pi_i)$, where $\pi_i = P(i_1 = s_i)$, $i = 1, 2, \ldots, N$ represents the probability of being in state $s_i$ at time $t = 1$.

4. State transition matrix $A$

$$A = \begin{bmatrix} a_{11} & a_{12} & \cdots & a_{1N} \\ a_{21} & a_{22} & \cdots & a_{2N} \\ \cdots & \cdots & \cdots & \cdots \\ a_{N1} & a_{N2} & \cdots & a_{NN} \end{bmatrix}, \tag{4}$$

Where $a_{ij} = P(i_{t+1} = s_j | i_t = s_i)$, $i = 1, 2, \ldots, N; j = 1, 2, \ldots, N$ represents the probability of transitioning from state $s_i$ at time $t$ to state $s_i$ at time $t + 1$, given that the system is in state $s_i$ at time $t$.

5. Observation probability matrix $B$

$$B = \begin{bmatrix} b_{11} & b_{12} & \cdots & b_{1M} \\ b_{21} & b_{22} & \cdots & b_{2M} \\ \cdots & \cdots & \cdots & \cdots \\ b_{N1} & b_{N2} & \cdots & b_{NM} \end{bmatrix}, \tag{5}$$

Where $b_{jk} = P(o_t = v_k | i_t = s_j)$, $k = 1, 2, \ldots, M; j = 1, 2, \ldots, N$ represents the probability of generating observation $v_k$ given that the system is in state $s_j$ at time $t$.

In the model construction phase, we use the learning problem of HMM to establish an HMM $\lambda = (A, B, \pi)$ that conforms to the normal state based on non-attack stage alert information. Since there are no multiple stages that match HMM states in the alert information of the normal system state, we cluster the vectorized non-attack stage alerts, with the clusters

analogized to the hidden states of the HMM. This is because similar alerts are often clustered into the same cluster, which can be viewed as a stage. After determining the hidden states, model parameters are estimated, and the MSP is constructed.

This paper assumes that the process of clustering alerts into different clusters in multi-stage attacks follows the assumption of homogenous Markovity. The homogenous Markovity assumption posits that the hidden Markov chain's state at any given time $t$ depends only on its state at the previous time step, independent of other time steps' states and observations, and also independent of time $t$, which can be expressed as:

$$P(s_t|s_{t-1}, a_{t-1}, \ldots, s_1, a_1) = P(s_t|s_{t-1}), t = 1, 2, 3, \ldots, k, \tag{6}$$

Where $s_t$ represents the state at time $t$.

If there are $k$ clusters for non-attack stage alerts, then the HMM has k hidden states, i.e., $\{s_1, s_2, \ldots, s_k\}$. The initial state probability vector describes the probabilities of the model being in different states initially, and the state transition matrix describes the probabilities of transitions between states. Calculating probabilities based on frequency, for the initial state probability vector $\pi = (\pi_1, \pi_2, \ldots, \pi_N)$, $\pi_i = d_i/D$, where $d_i$ represents the count of states being $i$ in the sequence, and $D$ represents the total count of the sequence.

**3.3.2 Determining the hidden states of HMM.** Basically, similar alerts are positioned close to each other in the vector space, making it possible to utilize clustering algorithms to assist HMM modeling. Clustering algorithms are widely employed in anomaly detection [34]. In our approach, we use clustering algorithms to group alerts with similar semantic representations into the same cluster, while alerts with significantly different semantic representations are dispersed into separate clusters. Based on the dataset's characteristics, we employ the K-Means algorithm for alert clustering. Initially, k cluster centers are randomly selected, and then, by minimizing the error function through computing the distance between samples and cluster centers, the cluster centers are iteratively updated until the algorithm converges. In our work, we use the Euclidean distance as the distance metric, and the specific calculation formula is as follows:

$$d_{ij} = (\sum_{k=1}^{m}|x_{ki} - \mu_{kj}|^2)^{\frac{1}{2}}, \tag{7}$$

Where $x_i$ represents the $i$-th sample, $\mu_j$ represents the $j$-th cluster center vector, and $m$ represents the dimension of the sample. The error function is defined as the square error.

$$E = \sum_{k=1}^{K} \sum_{x \in C_i} \| x - \mu_i \|_2^2, \tag{8}$$

Where $K$ is the number of clusters, and $C_i$ represents the $i$-th cluster.

However, one drawback of the K-Means algorithm is that it cannot automatically determine the optimal number of clusters. To address this issue, in this paper, we employ the average silhouette score method to select an appropriate number of clusters. The average silhouette score method combines intra-cluster cohesion and inter-cluster separation, providing a more accurate assessment of clustering quality. Its objective is to minimize the intra-cluster distance

while maximizing the inter-cluster distance.

$$S(i) = \frac{b(i) - a(i)}{\max\{a(i), b(i)\}} = \begin{cases} 1 - \frac{a(i)}{b(i)}, & a(i) < b(i) \\ 0, & a(i) = b(i) \\ \frac{b(i)}{a(i)} - 1, & a(i) > b(i) \end{cases} \tag{9}$$

Where $a(i)$ represents the average dissimilarity of sample $i$ to other points within the same cluster. Smaller $a(i)$ values indicate lower dissimilarity of sample $i$ within its cluster, suggesting that sample $i$ should belong more to that cluster. $b(i)$ represents the minimum average dissimilarity of sample $i$ to points in other clusters, where $b(i) = \min\{b_{i1}, b_{i2}, \ldots, b_{iK}\}$, with larger $b(i)$ values indicating that sample $i$ is less likely to belong to other clusters. After computing the silhouette score for all samples, taking the average yields the average silhouette score, which falls within the range of $[-1, 1]$. Larger silhouette scores indicate that samples within clusters are closer, while distances between samples from different clusters are greater, resulting in a better clustering effect.

### 3.4 Attack detection

**3.4.1 Threshold determination.** In the attack detection phase, our work begins by determining the decision threshold. For threshold determination, we make an assumption that if an alert is generated during an attack stage, it must exhibit differences from alerts during non-attack stages. In this case, we introduce the concept of Generative Probability (GP), which represents the strength of an alert's fit to the MSP. Algorithm 1 describes the process of using GP to determine the detection threshold. Firstly, the alert sequence in the attack stage is input into the trained MSP model. Then, GP is calculated through probability computation. The distribution of GP is statistically analyzed, and the average GP is calculated and set as the threshold.

**Algorithm 1**: Determining the Threshold

```
Require: Attack Phase Alert Sequence A = [a₁, a₂, ..., aₙ] ⊆ alerts
Ensure: Threshold Value θ
1: GP_count = {}, total_alerts = len(A)
2: for each value aᵢ in A do
3:    GPᵢ = MSP(aᵢ)
4:    if GPᵢ not in GP_count then
5:      GP_count[GPᵢ] = 1
6:    else
7:      GP_count[GPᵢ] + = 1
8:    end if
9: end for
10: θ = (∑ⁿᵢ₌₀(GPᵢ × GP_count[GPᵢ])) / total_alerts
11: return θ
```

**3.4.2 Attack determination.** We use the probability calculation problem from HMM's three problems to compute the GP of an alert fitting the MSP. When GP is greater than the set threshold, it indicates that the alert is likely to occur in the MSP, classifying it as a non-attack stage alert. When GP is less than the set threshold, it suggests that the alert is less likely to occur in the MSP, classifying it as an attack stage alert, indicating the presence of a multi-stage

attack.

$$Stage_i = \begin{cases} 0 & , \alpha_i \geq \theta \\ 1 & , \alpha_i < \theta \end{cases},$$

(10)

Where $\alpha_i$ represents the probability that sample $i$ fits the MSP, and $\theta$ is the set threshold. When $Stage_i = 0$, it indicates that sample $i$ is a non-attack stage alert. When $Stage_i = 1$, it signifies that sample $i$ is an attack stage alert.

## Section 4: Experiments and results analysis

In this section, we first introduced the model evaluation criteria in Section 4.1. Then, in Section 4.2, we provided an introduction to the datasets used in the experiments and described the multi-stage attacks included in them. Section 4.3 covered the hyperparameters involved in our experiments. Finally, in Section 4.4, we reported the results of our experiments.

### 4.1 Evaluation indicators

In the evaluation task, the most commonly used model evaluation metrics are accuracy, precision, recall, and F1-score. These metrics provide comprehensive insights into the model's performance, particularly in classification tasks. Therefore, this paper selects these four metrics as the model evaluation criteria, and these metrics are calculated using the following formulas.

$$accuracy = \frac{TP + TN}{TP + TN + FP + FN},$$

(11)

$$precision = \frac{TP}{TP + FP},$$

(12)

$$recall = \frac{TP}{TP + FN},$$

(13)

$$F1\_score = \frac{2 * precision * recall}{precision + recall},$$

(14)

Here, TP represents the number of attack stage alerts correctly identified as attacks, TN represents the number of non-attack stage alerts correctly identified as normal, FP represents the number of non-attack stage alerts incorrectly identified as attacks, and FN represents the number of attack stage alerts incorrectly identified as normal. Accuracy reflects the overall accuracy of identification, precision reflects the proportion of correctly identified attack alerts among those detected as attacks, recall reflects how many attack stage alerts were detected, and the F1-score is the harmonic mean of precision and recall, reflecting the model's overall performance.

### 4.2 Dataset introduction

The datasets relevant to multi-stage attack detection include DARPA2000, ISCXIDS2012, CTU-13, DEFCON21 CTF, and CICIDS2017, among others. In these datasets, DARPA2000 [35] contains specific scenarios with multi-stage attack samples, making it one of the most commonly used datasets in related research. Compared to two other datasets in the same series, DARPA1998 and DARPA1999, this dataset eliminates many generation errors and

**Table 2. Statistics of the alert data used in the experiments.**

| Dataset | Stage | Number of packets | Number of alerts | Label |
|---|---|---|---|---|
| DARPA2000 | s1 | 785 | 42 | 1 |
| | s2 | 148 | 2788 | 1 |
| | s3 | 530 | 65 | 1 |
| | s4 | 526 | 6 | 1 |
| | s5 | 34533 | 34846 | 1 |
| | o | 355547 | 139031 | 0 |
| ISCX2012 | s1 | 78333 | 124 | 1 |
| | s2 | 20582 | 528 | 1 |
| | s3 | 87900 | 41 | 1 |
| | s4 | 10103 | 525 | 1 |
| | s5 | 398621 | 464 | 1 |
| | s6 | 698149 | 323 | 1 |
| | o | 4469461 | 3726 | 0 |
| CICIDS2017 | s1 | 1507 | 741 | 1 |
| | s2 | 652 | 169 | 1 |
| | s3 | 79 | 96 | 1 |
| | o | 168186 | 1106 | 0 |

biases. It includes two attack scenarios, LLSDDOS1.0 and LLSDDOS2.0, with five attack stages. The ISCXIDS2012 [36] dataset consists of real network traffic and encompasses complete multi-stage attacks with six stages, making it a common dataset for multi-stage attack detection research. The CICIDS2017 [37] dataset is a supplement to the ISCXIDS2012 dataset and has also seen extensive use in recent related research. The data were collected continuously for five days from Monday to Friday and have been labeled.

Numerous related studies have conducted experimental validation on these three datasets, making them suitable for quantitative comparisons between different methods. Specifically, on the DARPA2000 dataset, we primarily used LLSDDOS1.0 because it generates a more significant number of alerts compared to LLSDDOS2.0, allowing for more comprehensive training and validation of our method. On the CICIDS2017 dataset, we selected the Thursday morning dataset, which includes three types of web attacks from the same attacker: Brute Force, Cross-Site Scripting (XSS) attacks, and SQL injection attacks, composing a multi-stage web attack scenario.

The detection method proposed in this paper takes alerts generated by the Snort intrusion detection system as input. Therefore, the original network data in pcap packet files needs to be converted into alert data through methods such as replay. In this paper, the three datasets were replayed using the Tcpreplay tool, and Snort v2.9.7(with default rules) was used to detect the replayed traffic data and generate alert data. During data preprocessing, alerts in the non-attack stages were labeled as 0, while alerts in the attack stages were labeled as 1. Table 2 displays the statistics of alert data used in our experiments.

## 4.3 Experimental settings

The experimental setup for this study is as follows:

First, the experiments were conducted on a Windows 10 platform with an Intel(R) Core (TM) i5-7300HQ CPU @ 2.50GHz processor and 8GB of RAM. The development environment utilized Python 3.8, Keras 2.4.0, and TensorFlow 2.3.0.

**Table 3. Hyperparameter configuration.**

| Hyperparameter | Value | Description |
| --- | --- | --- |
| dm | 1 | Set Doc2Vec to use the PV-DM model. |
| size | 200 | Sentence vector dimension. |
| windows | 10 | Window size. |
| negative | 10 | Negative sampling ratio. |
| niter | 10 | Maximum iteration number of HMM. |
| tol | 0.01 | HMM convergence threshold. |

Then, the network traffic data were replayed by Tcpreplay(version 4.2.6) and the alert texts were obtained by Snort(version 2.9.7) with default intrusion detection rules. The alert texts were interpreted with PV-DM by Doc2Vec before building MSP. Scikit-learn(version 1.0.2) and hmmlearn(version 0.2.3) were adopted in the process of building MSP. Hyper-parameters in the experiments are summarized in Table 3.

### 4.4 Analysis of experimental results

A series of experiments were conducted sequentially to effectively evaluate the method proposed in this paper, with each module being trained in turn. First, the alert pre-processing stage trains the alert vectorization model. Then, the MSP is constructed, and non-attack phase alert data undergo clustering. To obtain the optimal value of $k$, this paper uses the average silhouette coefficient method to select an appropriate $k$ value, as shown in Figs 4–6.

Fig 4 shows the silhouette coefficient distribution for clustering data in the DARPA2000 attack scenario with different values of k. Each small graph represents a clustering result, where the height of the shape indicates the number of alerts in the cluster, the width represents the silhouette coefficient arrangement of alerts in the cluster (wider is better), and the dashed line represents the average silhouette coefficient. Comparing k values from 2 to 9, it is observed that when k is equal to 3, the distribution of clusters is relatively even, and the silhouette

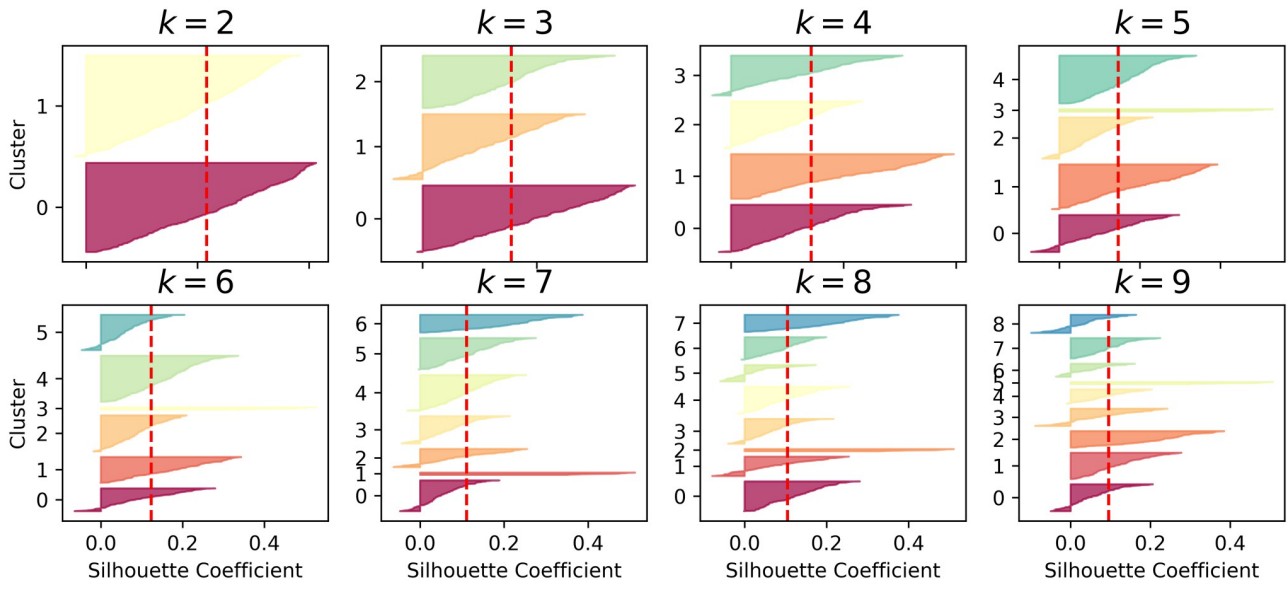

**Fig 4. Silhouette coefficient distribution for different k values in the DARPA2000 dataset.**

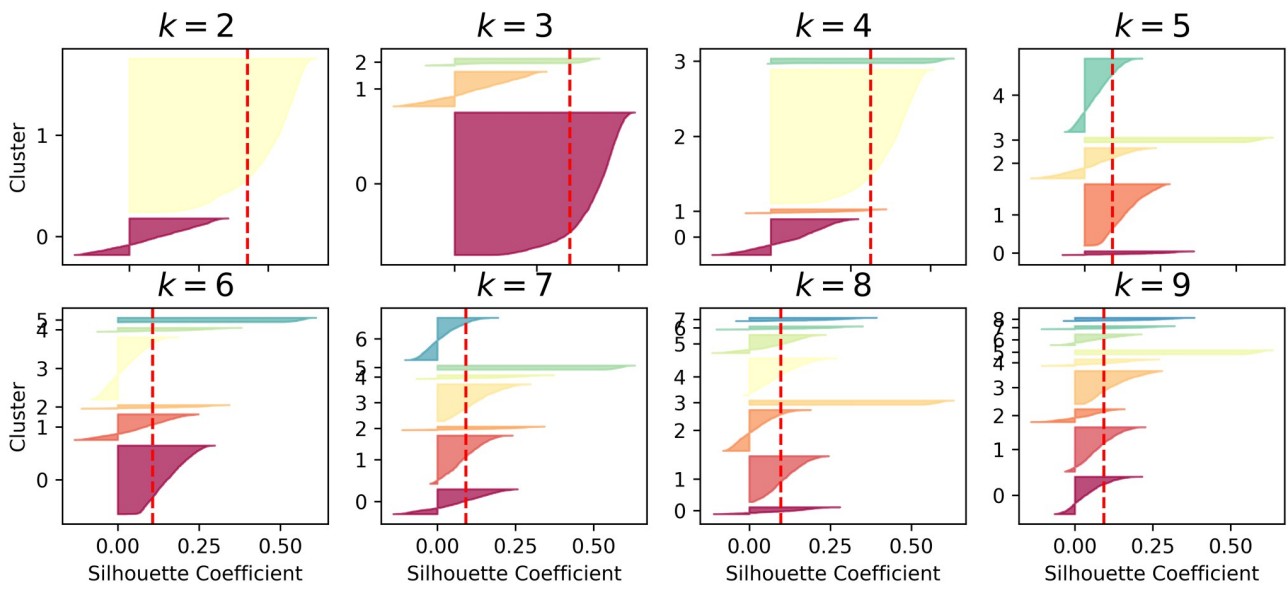

**Fig 5. Silhouette coefficient distribution for different k values in the ISCXIDS2012 dataset.**

coefficients for each cluster are above the average silhouette coefficient. Therefore, $k = 3$ is chosen as the number of clusters. Fig 5 displays the silhouette coefficient distribution for clustering data in the ISCXIDS2012 attack scenario with different k values. It is observed that when $k = 9$, the distribution of clusters is relatively even, making $k = 9$ the optimal number of clusters for the ISCXIDS2012 attack scenario. Fig 6 illustrates the silhouette coefficient distribution for clustering data in the CICIDS2017 attack scenario. For k values ranging from 3 to 9, some clusters have silhouette coefficients lower than the average silhouette coefficient, except for $k = 4$,

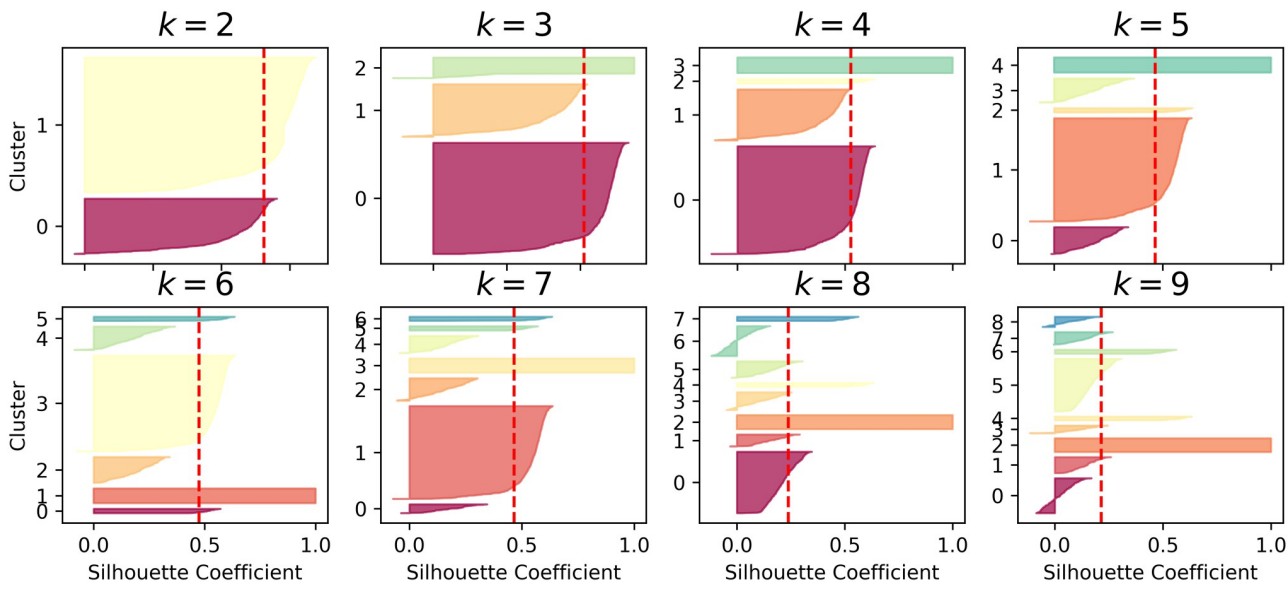

**Fig 6. Silhouette coefficient distribution for different k values in the CICIDS2012 dataset.**

**Table 4. k-Values selected for the three datasets.**

| Dataset | k-Value |
| --- | --- |
| DARPA2000 | 3 |
| ISCX2012 | 9 |
| CICIDS2017 | 4 |

where more clusters have silhouette coefficients extending to the right and approaching 1.0. Therefore, $k = 4$ is chosen as the optimal number of clusters for the CICIDS2017 dataset. The selected k values for all three datasets are summarized in Table 4:

After obtaining the optimal cluster number k using the silhouette coefficient method, a clustering model is built for non-attack phase alerts. Similar alerts are grouped into the same cluster, and the clustered sequence is used as the hidden state sequence to construct the MSP.

In Fig 7, it represents the GP distribution of alerts in the attack phase for the DARPA2000, ISCXIDS2012, and CICIDS2017 attack scenarios. In the DARPA2000 dataset, there are 37,747 attack phase alerts, of which 37,738 alerts have a GP of 0, accounting for 99.97% of the total alerts. Nine alerts have a GP of 0.005. According to Algorithm 1, the attack detection threshold for this attack scenario is set to 0. In the ISCXIDS2012 dataset, there are 2,005 attack phase alerts, and all GPs are 0. Therefore, the attack detection threshold is set to 0. In the CICIDS2017 dataset, there are 1,006 attack phase alerts, of which 998 alerts have a GP of 0, accounting for 99.20% of the total alerts. Two alerts have a GP of 0.001, and six alerts have a GP of 0.0236. According to Algorithm 1, the attack detection threshold for this attack scenario is set to 0.0001 (approximately 0). These results confirm our earlier assumption. Therefore, we set the attack detection threshold to 0, meaning that if the GP of an alert sequence is 0, it is labeled as an attack phase alert; otherwise, it is labeled as a non-attack phase alert.

Furthermore, to further validate the performance of our method, we compared it with several recent multi-stage detection methods [8, 14, 27]. Reference [8] introduced a multi-stage attack detection method that builds a sequence-to-sequence model using an encoder-decoder structure. Reference [14] proposed a detection method using pre-trained HMM. Reference [27] presented a detection method based on a multi-feature-layer Long Short-Term Memory (LSTM) network. For ease of reference, we will abbreviate these three methods as Seq2seq, Pre-HMM, and LSTM, respectively. The experiments were run on the three datasets, and the performance metrics are shown in Tables 5–7.

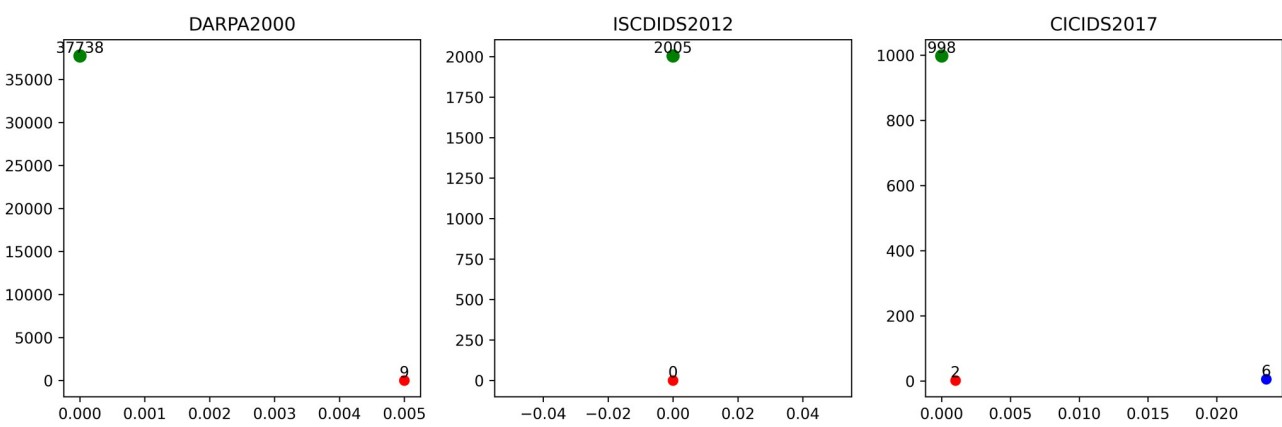

**Fig 7. The distribution of GP for attack phase alerts.**

**Table 5. Model evaluation results on the DARPA2000 dataset.**

| Method | Accuracy | Precision | Recall | F1-scores |
|---|---|---|---|---|
| Seq2seq | 0.985 | 0.936 | 0.952 | 0.930 |
| Pre-HMM | 0.852 | 0.795 | 0.813 | 0.806 |
| LSTM | 0.950 | 0.930 | 0.925 | 0.948 |
| **MSP** | **0.992** | **1.0** | **0.961** | **0.980** |

**Table 6. Model evaluation results on the ISCXIDS2012 dataset.**

| Method | Accuracy | Precision | Recall | F1-scores |
|---|---|---|---|---|
| Seq2seq | 0.987 | 0.994 | 0.936 | 0.979 |
| Pre-HMM | 0.625 | 0.594 | 0.659 | 0.609 |
| LSTM | 0.904 | 0.827 | 0.962 | 0.903 |
| **MSP** | **1.0** | **1.0** | **1.0** | **1.0** |

**Table 7. Model evaluation results on the CICIDS2012 dataset.**

| Method | Accuracy | Precision | Recall | F1-scores |
|---|---|---|---|---|
| Seq2seq | 0.976 | 0.995 | 0.938 | 0.952 |
| Pre-HMM | 0.826 | 0.804 | 0.832 | 0.873 |
| LSTM | 0.903 | 0.947 | 0.962 | 0.890 |
| **MSP** | **0.995** | **1.0** | **0.962** | **0.981** |

From Tables 5–7, it can be observed that on all three datasets, whether it is Accuracy, Precision, Recall, or the comprehensive F1-score, our method exhibited the best performance. On the DARPA2000 dataset, our method achieved 99.2% Accuracy and 100% Precision, with an F1-score of 98%, surpassing Pre-HMM, which also utilizes HMM. The deep learning methods Seq2Seq and LSTM had F1-scores of 93% and 94.8%, slightly lower than our approach. On the ISCXIDS2012 dataset, our method attained 100% across all four metrics. Seq2Seq and LSTM both had F1-scores above 90%, with Seq2Seq performing better. Pre-HMM still exhibited the poorest performance with an F1-score of only 60.9%. For the CICIDS2017 dataset, our method achieved an F1-score of 98.1%, followed by Seq2Seq at 95.2%, LSTM at 89%, and Pre-HMM at the lowest with 87.3%. According to our analysis, the main reasons for this result are as follows: First, we use doc2vec to vectorize the alarm text, which is more conducive to capturing the deep information of the alarm information and mining the internal correlation between each other, so the performance is obviously better than the Pre-HMM using word2vec; Secondly, our method is mainly to find abnormal conditions in the alarm sequence without specifically detecting the attack stage, so the accuracy is high. Lastly, multi-stage attack detection is not time-sensitive, which is why Seq2Seq outperforms LSTM on all datasets as a deep learning method.

Furthermore, to validate the analysis of performance overhead in different phases as discussed in Section 3.1, we conducted experiments and collected statistics on the time taken for model training and detection in each phase. The results are presented in Fig 8.

Fig 8 illustrates the performance overhead of the proposed method on different datasets. As analyzed in Section 3.1, the performance overhead of the method is mainly concentrated in

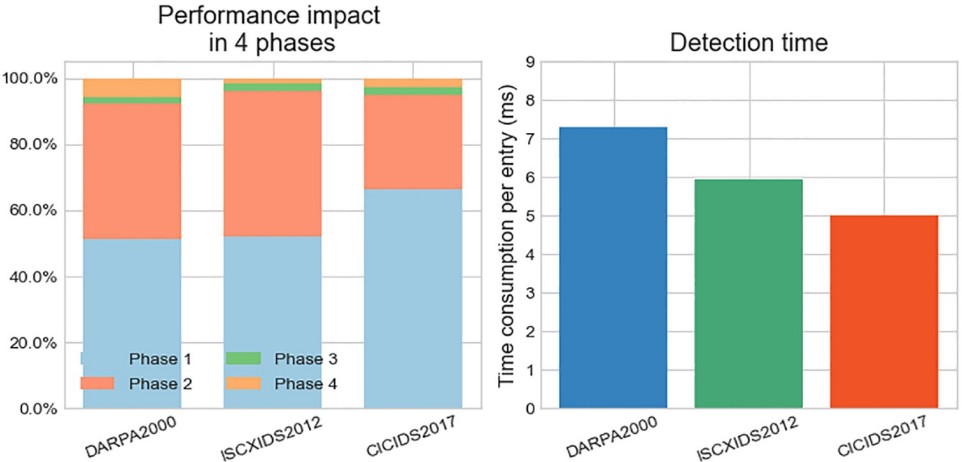

**Fig 8. The performance overhead of different datasets at each phase.**

the first three phases (phase1 to phase3), accounting for over 95% of the total performance overhead on all three datasets. In contrast, phase4, the detection phase, incurs a time cost of only 5%, 2%, and 3% of the overall time cost, respectively. The primary work in phase1 to phase3 is conducted offline, and therefore, it does not impact the detection performance of the proposed method. In phase4, the time required for detecting a single attack sequence is approximately 7ms on the DARPA2000 dataset, around 6ms on the ISCXIDS2012 dataset, and about 5ms on the CICIDS2017 dataset, almost achieving real-time detection.

## Section 5: Conclusion

### 5.1 Summary

In this paper, we present an anomaly-based multi-stage attack detection method. By modeling the normal state of a stable system and constructing an MSP, our aim is to detect attack behaviors. Our objective is to develop a model capable of detecting unknown pattern attacks, not just common ones like DoS. Our approach starts by vectorizing alert information to better capture the deep-seated information within alerts. Next, we process the vectorized data of non-attack stage alerts, using clustering and HMM to build the MSP. Finally, we perform detection on the alert data collected by IDS, using the alert's fit to the MSP's generated probability as the basis for judgment to determine if the alert belongs to an attack stage.

We conducted experiments using three publicly available datasets: DARPA2000, ISC-XIDS2012, and CICIDS2017, and compared our method with three state-of-the-art multi-stage attack detection methods: Seq2seq, Pre-HMM, and LSTM. The experimental results demonstrate that our method performs better in detecting alerts in attack stages. For the ISC-XIDS2012 dataset, our method can completely detect existing attack stage alerts, while achieving detection accuracy of over 99% on the other two datasets, with precision rates of 100%.

### 5.2 Future work

The method proposed in this paper can detect the presence of attack behavior and identify which alert information characterizes the attack. However, it cannot recognize the specific stage of the attack and reconstruct the attack scenario. Therefore, this method is not suitable

for extracting attack scenarios. In future work, we will focus on more refined attack stage detection and attack scenario extraction.

Furthermore, the selection of alert information generated by Snort has been a consistent methodology. However, it should be noted that Snort does not have the ability to detect all attack behaviors and generate alert information. The detection capability of Snort is highly dependent on the rules used. Therefore, in our future research, we will attempt to use different sets of Snort rules and consider their false positives in legitimate traffic.

In the research on multi-stage attack detection, most researchers have chosen Snort default rules to generate alert information for different datasets [8, 14, 18], making their methods reproducible and easy to compare. In fact, for network traffic, Snort's default rules can only capture a small portion of attacks (10%), and when additional rules are used, the proportion of captured attacks increases to 80%, but this also reduces the accuracy from nearly 100% with default rules to 1.5%. Hence, another task for our future endeavors is to develop more insightful Snort rules, aiming to enhance data quality and, consequently, improve detection accuracy.

## Author Contributions

**Conceptualization:** Wei Ma, Yunyun Hou, Mingyu Jin, Pengpeng Jian.

**Data curation:** Wei Ma, Yunyun Hou.

**Formal analysis:** Yunyun Hou, Mingyu Jin.

**Funding acquisition:** Wei Ma, Pengpeng Jian.

**Investigation:** Yunyun Hou, Mingyu Jin, Pengpeng Jian.

**Methodology:** Wei Ma, Yunyun Hou.

**Project administration:** Wei Ma.

**Software:** Yunyun Hou.

**Supervision:** Wei Ma.

**Validation:** Yunyun Hou, Mingyu Jin, Pengpeng Jian.

**Visualization:** Wei Ma, Yunyun Hou, Mingyu Jin, Pengpeng Jian.

**Writing – original draft:** Yunyun Hou.

**Writing – review & editing:** Wei Ma.

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
