## [Decision Letter · Decision Letter 0]

4 Oct 2023

PONE-D-23-29167Anomaly based multi-stage attack detection methodPLOS ONE

Dear Dr. Hou,

Thank you for submitting your manuscript to PLOS ONE. After careful consideration, we feel that it has merit but does not fully meet PLOS ONE’s publication criteria as it currently stands. Therefore, we invite you to submit a revised version of the manuscript that addresses the points raised during the review process.

We look forward to receiving your revised manuscript.

Kind regards,

Praveen Kumar Donta, Ph.D.

Academic Editor

PLOS ONE

Journal Requirements:

Reviewers' comments:

Reviewer's Responses to Questions

**Comments to the Author**

1. Is the manuscript technically sound, and do the data support the conclusions?

Reviewer #1: Yes

Reviewer #2: Partly

Reviewer #3: Yes

2. Has the statistical analysis been performed appropriately and rigorously? 

Reviewer #1: Yes

Reviewer #2: Yes

Reviewer #3: Yes

3. Have the authors made all data underlying the findings in their manuscript fully available?

Reviewer #1: Yes

Reviewer #2: No

Reviewer #3: Yes

4. Is the manuscript presented in an intelligible fashion and written in standard English?

Reviewer #1: Yes

Reviewer #2: Yes

Reviewer #3: Yes

5. Review Comments to the Author

Reviewer #1: The article is well written,

1. The main contribution and originality should be explained in more detail; why is it important?

2. More comparison of results against alternative approaches is needed for the benefit of the readers.

3. Authors should argue their choice of performance evaluation indicators.

4. The Introduction could be updated with recent reviews dedicated to the references related to the topic addressed. In literature study the highlighted article might be considered (https://doi.org/10.1007/s11042-023-15363-4) for inclusion.

5. The English language should be more polished, and some typo corrections are required.

6. There are different algorithm approaches currently in use. Why the author's selection towards the proposed system is preferred.

7. Rewrite or give reasonable explanations of notations used in equations.

8. List the limitations of the proposed system.

9. Describe the weakness of the proposed system.

Reviewer #2: The authors propose a novel and promising approach to accurately identifying multi-stage attacks in cyberspace. The method constructs a Multi-Stage Profile (MSP) by modeling the stable system's normal state and employs Doc2Vec and Hidden Markov Models (HMM) to detect attack behaviors. The comparative experiments were conducted on three different datasets, namely DARPA2000, ISCXIDS2012, and CIC-IDS2017. The experimental results demonstrate that this method achieves an over 99% accuracy and a 100% precision in multi-stage attack detection.

However, I recommend revising the paper before proceeding further. References are shown as question marks (e.g., [?] on line 20 of page 2) and figures are numbered as "???" (e.g., Fig ?? on lines 35-36 of page 2). This makes it very difficult to understand which figures and references are being cited in the text.

Reviewer #3: In this paper, an anomaly-based multi-stage attack detection method is proposed. Designed experiments showed the efficiency and effectiveness of the proposed methods. The investigated issue is interesting, and the paper shows the research depth and expertise. However, minor revision is required as follows.

1. The key novel contribution in the proposed framework in Figure 2 should be well described, it looks like authors integrate the recent techniques together to form the framework.

2. The performance impact of each stage in Figure 2 can be introduced and discussed.

3. For the evaluation, the dataset/algorithm choosing reasons, the detailed platform configurations and the discussion on other untested datasets should be introduced in the revised paper.

4. Some references lack the necessary information (e.g., [2]), please provide all information according to the right template.

5. A notation table can be add to explain all math symbols.

6. Please go through the paper carefully and double check whether the right template are used. Correct some typos and formatting issues (e.g., why all citation and figure refs do not display correctly).

7. Make the References more comprehensive, besides this work, some other promising scenarios (e.g., Big data, other IoT systems) can be covered in this work. If the above related work can be discussed, it can strongly improve the research significance. For the improvement, the following papers can be considered to make the references more comprehensive.

Jin Wang, Changqing Zhao, Shiming He, Yu Gu, Osama Alfarraj, Ahed Abugabah, LogUAD: Log Unsupervised Anomaly Detection Based on Word2Vec, Computer Systems Science and Engineering, 2022, 41(3): 1207–1222

Kai Zhong, ZhiBang Yang, Guoqing Xiao, Xingpei Li, Wangdong Yang, Kenli Li: An Efficient Parallel Reinforcement Learning Approach to Cross-Layer Defense Mechanism in Industrial Control Systems. IEEE Trans. Parallel Distributed Syst. 33(11): 2979-2990 (2022)

J. Wang, Y. Yang, T. Wang, R. Sherratt, J. Zhang. Big Data Service Architecture: A Survey. Journal of Internet Technology, 2020, 21(2): 393-405

J. Zhang, R. Zhang, O. Alfarraj, A. Tolba, G.-J. Kim. A Memory-Aware Spark Cache Replacement Strategy. Journal of Internet Technology, 2022, 23(6): 1185-1190

Bin Pu, Kenli Li, Shengli Li, Ningbo Zhu: Automatic Fetal Ultrasound Standard Plane Recognition Based on Deep Learning and IIoT. IEEE Trans. Ind. Informatics 17(11): 7771-7780 (2021)

Lixia Ji, Xiao Zhang, Yao Zhao, Zongkun Li, "Anomaly Detection of Dam Monitoring Data based on Improved Spectral Clustering," Journal of Internet Technology, vol. 23, no. 4 , pp. 749-759, Jul. 2022.

6. PLOS authors have the option to publish the peer review history of their article (what does this mean?). If published, this will include your full peer review and any attached files.

Reviewer #1: **Yes: **CHANDRAMOHAN D

Reviewer #2: No

Reviewer #3: No

---

## [Author Response · Author response to Decision Letter 0]

2 Nov 2023

Reviewer#1, Concern # 1: The main contribution and originality should be explained in more detail; why is it important? 

Author response: Thank you very much for your valuable comments. We have modified the manuscript to describe the specific contributions in detail in the first section, highlighting the innovations of this paper.

…

Compared to previous methods, our approach can extract deep-seated information from alerts, analyze the relationships between alert entries, and accomplish model building without the need for prior knowledge using unsupervised clustering methods.

The main contributions of this paper are as follows:

An original concept of Multi-Stage Profile (MSP) was introduced. Borrowing from the idea of dividing attacks into multiple stages, the paper performs staged modeling of a system's normal state. This enhances the model's adaptability and generalization to different network environments and attack types, improving its overall adaptability and generalization.

To accommodate various attack scenarios, a novel method for adaptive acquisition stages is proposed. This method utilizes Doc2Vec to deeply extract information from alert messages, preserving the interrelations between alerts more effectively. Subsequently, it clusters vectorized alert information, grouping semantically similar alerts into the same stage. This method demonstrates adaptability to various attack scenarios.

An original concept of Generating Probability (GP) is introduced, used to calculate the probability of an alert sequence occurring. We employ GP to establish anomaly thresholds, enabling attack detection.

Comparative experiments conducted on three different datasets (DARPA2000, ISCXIDS2012, CIC-IDS2017): Extensive experimental results illustrate that this method achieves an accuracy of over 99\\% in detecting multi-stage attacks across various datasets, outperforming the latest methods.

Reviewer#1, Concern # 2: More comparison of results against alternative approaches is needed for the benefit of the readers. 

Author response: Thank you very much for your valuable comments. Some more detailed analyses of the results are given in the Analysis of experimental results section of the manuscript to express that the performance of our method is better than the alternative approaches. And we also made a discussion to explain the reasons behind the results. The modifications are as follows:

…

From Table5, 6, and 7, it can be observed that on all three datasets, whether it is Accuracy, Precision, Recall, or the comprehensive F1-score, our method exhibited the best performance. On the DARPA2000 dataset, our method achieved 99.2% Accuracy and 100\\% Precision, with an F1-score of 98%, surpassing Pre-HMM, which also utilizes HMM. The deep learning methods Seq2Seq and LSTM had F1-scores of 93% and 94.8%, slightly lower than our approach. On the ISCXIDS2012 dataset, our method attained 100\\% across all four metrics. Seq2Seq and LSTM both had F1-scores above 90\\%, with Seq2Seq performing better. Pre-HMM still exhibited the poorest performance with an F1-score of only 60.9%. For the CICIDS2017 dataset, our method achieved an F1-score of 98.1%, followed by Seq2Seq at 95.2%, LSTM at 89%, and Pre-HMM at the lowest with 87.3%. According to our analysis, the main reasons for this result are as follows: First, we use doc2vec to vectorize the alarm text, which is more conducive to capturing the deep information of the alarm information and mining the internal correlation between each other, so the performance is obviously better than the Pre-HMM using word2vec; Secondly, our method is mainly to find abnormal conditions in the alarm sequence without specifically detecting the attack stage, so the accuracy is high. Lastly, multi-stage attack detection is not time-sensitive, which is why Seq2Seq outperforms LSTM on all datasets as a deep learning method.

Reviewer#1, Concern # 3: Authors should argue their choice of performance evaluation indicators. 

Author response: Thank you very much for your valuable comments. We have explained the reasons for selecting the evaluation criteria in the Evaluation Indicators section, and the specific changes are as follows:

…

In the evaluation task, the most commonly used model evaluation metrics are accuracy, precision, recall, and F1-score. These metrics provide comprehensive insights into the model's performance, particularly in classification tasks. Therefore, this paper selects these four metrics as the model evaluation criteria, and these metrics are calculated using the following formulas.

Reviewer#1, Concern # 4: The Introduction could be updated with recent reviews dedicated to the references related to the topic addressed. In literature study the highlighted article might be considered (https://doi.org/10.1007/s11042-023-15363-4) for inclusion. 

Author response: Thank you very much for your valuable comments. We added this literature (reference [10]), which is cited in the Introduction and Related Works sections, which is changed as follows.

…

Algorithms employed include Hidden Markov Model (HMM), Support Vector Machine, Decision Trees, Bayesian Networks, and Deep Neural Networks, among others [10].

…

Deep learning approaches can overcome some limitations of traditional shallow machine learning, capturing deep-seated features within the data, and enhancing detection performance [10].

[10] Dhasarathan C, Shanmugam M, Kumar M, Tripathi D, Khapre S, Shankar A. A nomadic multi-agent based privacy metrics for e-health care: a deep learning approach. Multimedia Tools and Applications. 2023; p. 1–24

Reviewer#1, Concern # 5: The English language should be more polished, and some typo corrections are required. 

Author response: Thank you very much for your valuable comments. We greatly appreciate your feedback on the language of the article. We have carefully considered your suggestions to ensure a smoother and more polished language. Additionally, we have thoroughly reviewed the article and made necessary corrections for spelling errors to ensure a high-quality publication.

Reviewer#1, Concern # 6: There are different algorithm approaches currently in use. Why the author's selection towards the proposed system is preferred. 

Author response: Thank you very much for your valuable comments. We believe that Hidden Markov Models (HMM) have extensive applications in intrusion detection, and, in comparison to other methods, HMM excels in its capacity to effectively model the stages. The alarm sequences align well with the HMM representation, as elaborated in the 'HMM' section of the 'Multi-stage profile' part of the manuscript. The details are as follows:

…

HMM is widely applied in attack detection, and literature[18-22] has provided substantial evidence of the effectiveness of HMM in attack detection. HMM has unique advantages in stage modeling, as alert sequence data can be effectively matched with the HMM representation. Therefore, this paper constructs MSP based on HMM.

Reviewer#1, Concern # 7: Rewrite or give reasonable explanations of notations used in equations. 

Author response: Thank you very much for your valuable comments. In accordance with your advice, we have added Table 1, which serves as a symbol table to explain the formula symbols used in the paper. The specific details are reflected in the manuscript.

Reviewer#1, Concern # 8: List the limitations of the proposed system. 

Author response: Thank you very much for your valuable comments. We have added a Future work section in the Conclusion section to explain the limitations of our proposed method. The details are as follows:

‘’‘

The method proposed in this paper can detect the presence of attack behavior and identify which alert information characterizes the attack. However, it cannot recognize the specific stage of the attack and reconstruct the attack scenario. Therefore, this method is not suitable for extracting attack scenarios. In future work, we will focus on more refined attack stage detection and attack scenario extraction.

Reviewer#1, Concern # 9: Describe the weakness of the proposed system. 

Author response: Thank you very much for your valuable comments. We have added a Future work section in the Conclusion section to explain the limitations and weaknesses of our proposed method, as well as future research directions. The details are as follows:

…

The method proposed in this paper can detect the presence of attack behavior and identify which alert information characterizes the attack. However, it cannot recognize the specific stage of the attack and reconstruct the attack scenario. Therefore, this method is not suitable for extracting attack scenarios. In future work, we will focus on more refined attack stage detection and attack scenario extraction.

Reviewer#2, Concern # 1: However, I recommend revising the paper before proceeding further. References are shown as question marks (e.g., [?] on line 20 of page 2) and figures are numbered as "???" (e.g., Fig ?? on lines 35-36 of page 2). This makes it very difficult to understand which figures and references are being cited in the text. 

Author response: Thank you very much for your valuable comments. We have reviewed our manuscript and corrected the issue of incorrect references, images, and table citations. The .tex file was compiled with latex miktex version 4.80 and the compiled .pdf file was checked with Microsoft Edge version 118.0 to make sure that no questions mark (which means unreachable reference) exists. We will submit the compiled PDF file for review when submitting the revised manuscript.

Reviewer#3, Concern #1: The key novel contribution in the proposed framework in Figure 2 should be well described, it looks like authors integrate the recent techniques together to form the framework. 

Author response: Thank you very much for your valuable comments. We have rewritten the description of the framework proposed in Figure 2 and the main innovations are emphasized. The specific details are as follows:

…

The overall architecture of the method proposed in this paper, as shown in Fig 2, consists of four steps: network data acquisition (phase 1), alert preprocessing (phase 2), establishment of MSP (phase 3), and attack detection (phase 4).

Firstly, in phase 1, First, in Phase 1, we designed an automated data acquisition method to obtain alert information from network traffic. Deployed IDS in the network continuously analyze traffic data captured from the network environment and generate alerts when suspicious packets are detected based on predefined rules. IDS may not detect complete multi-stage attacks, but when attackers attempt to infiltrate through multiple attack stages, IDS may capture individual attack actions and issue corresponding alerts. In Phase 1, the system primarily faced performance pressure stemming from IDS traffic analysis, and therefore, we adopted an offline analysis strategy to avoid impacting the overall system performance. 

However, it is worth noting that IDS systems generate a significant number of false positives. These alerts result from the inability of the alert generation rules to distinctly differentiate between normal and malicious activities within the network, and thus, do not represent genuine security threats[28]. Nevertheless, these non-attack stage alerts often carry information about the system's activity patterns, serving as a means to describe the system's normal state. Alerts generated by IDS are stored in the Alerts Database, which encompasses alert data from the system's normal state (referred to as non-attack stage alerts) and alert data from multi-stage attack states (referred to as attack stage alerts in this paper).

In phase 2, we introduce a method for alert preprocessing with the goal of transforming the text-style alert data generated in Phase 1 into data that can be used by machine learning algorithms. Since HMM cannot directly process alert information, we convert the alert data from the Alerts Database into vectors using the Doc2Vec algorithm. This allows us to extract deep-seated information from the alerts and analyze the associations between alert entries for further processing. The performance overhead in Phase 2 primarily stems from the training of the Doc2Vec model. Therefore, similar to Phase 1, we adopt an offline training strategy in Phase 2.

Then, the proposed MSP is constructed in Phase 3. We initially use a clustering approach to automatically obtain the stage division of normal alert vectors, which is then mapped to hidden states in the HMM to complete the construction of MSP based on HMM. Training the clustering model and HMM introduces a significant performance overhead in this phase. Therefore, similar to previous phases, we also employ an offline training approach.

Finally, in Phase 4, we perform online detection of alert data using the constructed MSP. The probability generated by the MSP is used as the basis for determining anomalies. This probability is compared to a predefined threshold to decide whether the sequence is anomalous. If it is, then the alert is marked as an attack stage alert. In contrast to Phases 1, 2, and 3, the detection process in Phase 4 is conducted online, using the MSP model obtained in the offline training of Phase 3. Online detection is nearly real-time, and it imposes relatively low performance overhead.

Reviewer#3, Concern #2: The performance impact of each stage in Figure 2 can be introduced and discussed. 

Author response: Thank you very much for your valuable comments. The performance impact of each phase shown in Figure 2 are discussed in the modified manuscript. And we also conducted a series of experiments to validate our analysis, with the results illustrated in Figure 8. The specific details are reflected in the manuscript.

Reviewer#3, Concern #3: For the evaluation, the dataset/algorithm choosing reasons, the detailed platform configurations and the discussion on other untested datasets should be introduced in the revised paper. 

Author response: Thank you very much for your valuable comments. We introduced the reasons for selecting datasets and algorithms, specific platform configurations, and reasons for not selecting other datasets in our paper. The details are as follows:

…

The datasets relevant to multi-stage attack detection include DARPA2000, ISCXIDS2012, CTU-13, DEFCON21 CTF, and CICIDS2017, among others. In these datasets, DARPA2000[35] contains specific scenarios with multi-stage attack samples, making it one of the most commonly used datasets in related research. Compared to two other datasets in the same series, DARPA1998 and DARPA1999, this dataset eliminates many generation errors and biases. It includes two attack scenarios, LLSDDOS1.0 and LLSDDOS2.0, with five attack stages. The ISCXIDS2012[36] dataset consists of real network traffic and encompasses complete multi-stage attacks with six stages, making it a common dataset for multi-stage attack detection research. The CICIDS2017[37] dataset is a supplement to the ISCXIDS2012 dataset and has also seen extensive use in recent related research. The data were collected continuously for five days from Monday to Friday and have been labeled.

…

The experimental setup for this study is as follows:

First, the experiments were conducted on a Windows 10 platform with an Intel(R) Core(TM) i5-7300HQ CPU @ 2.50GHz processor and 8GB of RAM. The development environment utilized Python 3.8, Keras 2.4.0, and TensorFlow 2.3.0.

Then, the network traffic data were replayed by Tcpreplay(version 4.2.6) and the alert texts were obtained by Snort(version 2.9.7) with default intrusion detection rules. The alert texts were interpreted with PV-DM by Doc2Vec before building MSP. Scikit-learn(version 1.0.2) and hmmlearn(version 0.2.3) were adopted in the process of building MSP. Hyper-parameters in the experiments are summarized in table 3.

…

HMM is widely applied in attack detection, and literature[18-22] has provided substantial evidence of the effectiveness of HMM in attack detection. HMM has unique advantages in stage modeling, as alert sequence data can be effectively matched with the HMM representation. Therefore, this paper constructs MSP based on HMM.

Reviewer#3, Concern #4: Some references lack the necessary information (e.g., [2]), please provide all information according to the right template. 

Author response: Thank you very much for your valuable comments. We have reviewed our manuscript and made modifications according to the format required by the journal.

Reviewer#3, Concern #5: A notation table can be add to explain all math symbols. 

Author response: Thank you very much for your valuable comments. In accordance with your advice, we have added Table 1, which serves as a symbol table to explain the formula symbols used in the paper. The specific details are reflected in the manuscript.

Reviewer#3, Concern #6: Please go through the paper carefully and double check whether the right template are used. Correct some typos and formatting issues (e.g., why all citation and figure refs do not display correctly). 

Author response: Thank you very much for your valuable comments. We have reviewed our manuscript and corrected the issue of incorrect references, images, and table citations. We will submit the compiled PDF file for review when submitting the revised manuscript.

Reviewer#3, Concern #7: Make the References more comprehensive, besides this work, some other promising scenarios (e.g., Big data, other IoT systems) can be covered in this work. If the above related work can be discussed, it can strongly improve the research significance. For the improvement, the following papers can be considered to make the references more comprehensive. 

Jin Wang, Changqing Zhao, Shiming He, Yu Gu, Osama Alfarraj, Ahed Abugabah, LogUAD: Log Unsupervised Anomaly Detection Based on Word2Vec, Computer Systems Science and Engineering, 2022, 41(3): 1207–1222

Kai Zhong, ZhiBang Yang, Guoqing Xiao, Xingpei Li, Wangdong Yang, Kenli Li: An Efficient Parallel Reinforcement Learning Approach to Cross-Layer Defense Mechanism in Industrial Control Systems. IEEE Trans. Parallel Distributed Syst. 33(11): 2979-2990 (2022)

J. Wang, Y. Yang, T. Wang, R. Sherratt, J. Zhang. Big Data Service Architecture: A Survey. Journal of Internet Technology, 2020, 21(2): 393-405

J. Zhang, R. Zhang, O. Alfarraj, A. Tolba, G.-J. Kim. A Memory-Aware Spark Cache Replacement Strategy. Journal of Internet Technology, 2022, 23(6): 1185-1190

Bin Pu, Kenli Li, Shengli Li, Ningbo Zhu: Automatic Fetal Ultrasound Standard Plane Recognition Based on Deep Learning and IIoT. IEEE Trans. Ind. Informatics 17(11): 7771-7780 (2021)

Lixia Ji, Xiao Zhang, Yao Zhao, Zongkun Li, "Anomaly Detection of Dam Monitoring Data based on Improved Spectral Clustering," Journal of Internet Technology, vol. 23, no. 4 , pp. 749-759, Jul. 2022.

Author response: Thank you very much for your valuable comments. We have carefully considered your suggestion and cited four of the six references you proposed (references [23] [24] [32] [34]). The specific details are as follows:

…

Furthermore, With the advancement of big data technology [23], some deep learning methods have been applied to multi-stage attack detection. Deep learning approaches can overcome some limitations of traditional shallow machine learning, capturing deep-seated features within the data [24], and enhancing detection performance.

[23] Wang J, Yang Y, Wang T, Sherratt RS, Zhang J. Big data service architecture: a survey. Journal of Internet Technology. 2020;21(2):393–405.

[24] Pu B, Li K, Li S, Zhu N. Automatic fetal ultrasound standard plane recognition based on deep learning and IIoT. IEEE Transactions on Industrial Informatics. 2021;17(11):7771–7780.

…

Word2Vec [32] addresses the dimensionality issue but loses sequence information by averaging word vectors. When using Word2Vec to compute text similarity, keyword extraction algorithms may not perform accurately.

[32] Wang J, Zhao C, He S, Gu Y, Alfarraj O, Abugabah A. LogUAD: log unsupervised anomaly detection based on Word2Vec. Computer Systems Science and Engineering. 2022;41(3):1207.

…

Basically, similar alerts are positioned close to each other in the vector space, making it possible to utilize clustering algorithms to assist HMM modeling. Clustering algorithms are widely employed in anomaly detection [34].

[34] Ji L, Zhang X, Zhao Y, Li Z. Anomaly Detection of Dam Monitoring Data based on Improved Spectral Clustering. Journal of Internet Technology. 2022;23(4):749–759

---

## [Decision Letter · Decision Letter 1]

6 Dec 2023

PONE-D-23-29167R1Anomaly based multi-stage attack detection methodPLOS ONE

Dear Dr. Hou,

Thank you for submitting your manuscript to PLOS ONE. After careful consideration, we feel that it has merit but does not fully meet PLOS ONE’s publication criteria as it currently stands. Therefore, we invite you to submit a revised version of the manuscript that addresses the points raised during the review process.

We look forward to receiving your revised manuscript.

Kind regards,

Praveen Kumar Donta, Ph.D.

Academic Editor

PLOS ONE

Reviewers' comments:

Reviewer's Responses to Questions

**Comments to the Author**

1. If the authors have adequately addressed your comments raised in a previous round of review and you feel that this manuscript is now acceptable for publication, you may indicate that here to bypass the “Comments to the Author” section, enter your conflict of interest statement in the “Confidential to Editor” section, and submit your "Accept" recommendation.

Reviewer #3: All comments have been addressed

Reviewer #4: (No Response)

Reviewer #5: All comments have been addressed

2. Is the manuscript technically sound, and do the data support the conclusions?

Reviewer #3: Yes

Reviewer #4: Partly

Reviewer #5: Yes

3. Has the statistical analysis been performed appropriately and rigorously? 

Reviewer #3: Yes

Reviewer #4: No

Reviewer #5: Yes

4. Have the authors made all data underlying the findings in their manuscript fully available?

Reviewer #3: Yes

Reviewer #4: No

Reviewer #5: Yes

5. Is the manuscript presented in an intelligible fashion and written in standard English?

Reviewer #3: Yes

Reviewer #4: Yes

Reviewer #5: Yes

6. Review Comments to the Author

Reviewer #3: The authors have very carefully addressed all my questions and comments.

Revised paper is much better and can be accepted now.

Reviewer #4: There are a lot of assumption That cannot be accepted.Results depens on the ruleset Used in snort, which it ¡S not identified in the paper. Only a Little part of multistage atacks can be detected Withnetwork traffic(around 30% of mittre att&ck attack matrix attacks). There are so many attacks that are not derected in snort rules (depending on ruleset configuration)

Reviewer #5: (No Response)

7. PLOS authors have the option to publish the peer review history of their article (what does this mean?). If published, this will include your full peer review and any attached files.

Reviewer #3: No

Reviewer #4: No

Reviewer #5: **Yes: **Monika Roopak

---

## [Author Response · Author response to Decision Letter 1]

14 Dec 2023

Reviewer#4, Concern # 1: The statistical analysis has not been conducted appropriately and rigorously.

Author response: Thank you very much for your valuable comments. We primarily employ the statistical method of 'representing probabilities in terms of frequencies.' This involves using statistical frequencies to substitute for Hidden Markov Model (HMM) initial probabilities and transition probabilities. This is a commonly used data statistical technique, and similar applications can be found in other literature. For instance, Zhang et al. [1] and Shawly et al. [2] have both utilized this approach in their respective works.

Reviewer#4, Concern # 2: The author has not provided all the data on which the manuscript relies.

Author response: Thank you very much for your valuable comments. All the data used in this paper have been sourced from public datasets, and the download link for the dataset has been provided in the references and on the Plos One submission page. Readers have the option to download the dataset from the website and replay the downloaded .pcap data through Tcpreplay 4.2.6. Snort 2.9.7 (utilizing the default Snort rules) has been employed to detect the replayed traffic and generate alarm data.

Reviewer#4, Concern # 3: There are a lot of assumption that cannot be accepted. Results depends on the ruleset used in Snort, which it is not identified in the paper. Only a Little part of multistage attacks can be detected With network traffic(around 30% of mittre att&ck attack matrix attacks). There are so many attacks that are not detected in Snort rules (depending on ruleset configuration).

Author response: Thank you for your valuable comments while reviewing my paper. The following are detailed answers to your questions.

First of all, the hypotheses we put forward in the paper are as follows:

1. If the contexts described by two alarms are similar, then they can be considered to have similar semantics.

2. The process of multi-stage attack from alarm clustering to each cluster conforms to homogeneous Markov property.

For the first hypothesis, we are based on the "context similarity principle", which means that similar contexts correspond to similar meanings. This assumption has also been used in literature [3]. For the second assumption, clusters after alarm clustering have similar behavior patterns or characteristics, and these clusters can be regarded as different states in the system, and this similarity is similar to the similarity between adjacent states in the Markov chain, which has the same assumption in literature [1]. These assumptions above are made mainly to simplify the model and make the experiment feasible.

Secondly, the Snort rules used in the experiments of this paper are all default rules, and we have added "default rules were used" in the manuscript.

Finally, you mentioned that only a small number of multi-stage attacks would be detected by Snort. However, in order to eliminate the impact of different Snort rules on detection and enhance the reproducibility and comparability of research, the methodology of multi-stage attack detection research has always used the alarm data generated by Snort default rules as the detection data source. For example, "Applications of Hidden Markov Models to Detecting Multi-stage Network Attacks" proposed by Ourston et al. [4] in 2003, In 2005 Mathew et al. [5] proposed "Real-time multistage attack awareness through enhanced intrusion alert clustering", "Real Time Alert Correlation and Prediction using Bayesian Networks" proposed by Ramaki et al. [6] in 2015, "Real-time multistep attack prediction based on Hidden Markov Models" proposed by Holgado et al. [7] in 2020, In 2020, Shawly et al. [2] proposed "Evaluation of HMM-Based Network Intrusion Detection System for Multiple Multi-Stage Attacks", "Multi-Stage Attack Detection via Kill Chain State Machines" proposed by Wilkens et al. [8] in 2021, In 2022, Zhang et al. [1] proposed "Multi-Step Attack Detection Based on Pre-Trained Hidden Markov Models", etc.

Thank you again for your modification suggestions. At the same time, writing more insightful Snort rules is also a direction of our future research, and we will focus on improving the detection effect in terms of data source quality in the future research.

References

[1] Zhang, X., Wu, T., Zheng, Q., Zhai, L., Hu, H., Yin, W., ... & Cheng, C. (2022). Multi-step attack detection based on pre-trained hidden Markov models. Sensors, 22(8), 2874.

[2] Shawly, T., Khayat, M., Elghariani, A., & Ghafoor, A. (2020). Evaluation of hmm-based network intrusion detection system for multiple multi-stage attacks. IEEE Network, 34(3), 240-248.

[3] Zhou, P., Zhou, G., Wu, D., & Fei, M. (2021). Detecting multi-stage attacks using sequence-to-sequence model. Computers & Security, 105, 102203.

[4] Ourston, D., Matzner, S., Stump, W., & Hopkins, B. (2003, January). Applications of hidden markov models to detecting multi-stage network attacks. In 36th Annual Hawaii International Conference on System Sciences, 2003. Proceedings of the (pp. 10-pp). IEEE. 

[5] Mathew, S., Britt, D., Giomundo, R., Upadhyaya, S., Sudit, M., & Stotz, A. (2005, October). Real-time multistage attack awareness through enhanced intrusion alert clustering. In MILCOM 2005-2005 IEEE Military Communications Conference (pp. 1801-1806). IEEE.

[6] Ramaki, A. A., Khosravi-Farmad, M., & Bafghi, A. G. (2015, September). Real time alert correlation and prediction using Bayesian networks. In 2015 12th International Iranian Society of Cryptology Conference on Information Security and Cryptology (ISCISC) (pp. 98-103). IEEE.

[7] Holgado, P., Villagrá, V. A., & Vazquez, L. (2017). Real-time multistep attack prediction based on hidden markov models. IEEE Transactions on Dependable and Secure Computing, 17(1), 134-147.

---

## [Decision Letter · Decision Letter 2]

27 Feb 2024

PONE-D-23-29167R2Anomaly based multi-stage attack detection methodPLOS ONE

Dear Dr. Hou,

Thank you for submitting your manuscript to PLOS ONE. After careful consideration, we feel that it has merit but does not fully meet PLOS ONE’s publication criteria as it currently stands. Therefore, we invite you to submit a revised version of the manuscript that addresses the points raised during the review process.

We look forward to receiving your revised manuscript.

Kind regards,

Praveen Kumar Donta, Ph.D.

Academic Editor

PLOS ONE

Journal Requirements:

Additional Editor Comments:

Please consider the reviewer comments, and resubmit your manuscript after care address of his/her comments.

Reviewers' comments:

Reviewer's Responses to Questions

**Comments to the Author**

1. If the authors have adequately addressed your comments raised in a previous round of review and you feel that this manuscript is now acceptable for publication, you may indicate that here to bypass the “Comments to the Author” section, enter your conflict of interest statement in the “Confidential to Editor” section, and submit your "Accept" recommendation.

Reviewer #4: All comments have been addressed

2. Is the manuscript technically sound, and do the data support the conclusions?

Reviewer #4: Partly

3. Has the statistical analysis been performed appropriately and rigorously? 

Reviewer #4: Yes

4. Have the authors made all data underlying the findings in their manuscript fully available?

Reviewer #4: Yes

5. Is the manuscript presented in an intelligible fashion and written in standard English?

Reviewer #4: Yes

6. Review Comments to the Author

Reviewer #4: The work has improved in this latest revision. In their response, the authors refer to the cited paper [3], "Detecting multi-stage attacks using sequence-to-sequence model," to justify the assumption that all stages of an attack are captured by the IDS. However, in the cited article, it is assumed that: "The baseline IDS is robust enough to capture sufficient alerts for representing any potential attacking stages. We can focus on the design of sequence-to-sequence detection without worrying about the input sequence may not reflect the attacking stages. We acknowledge this assumption may be too strong, especially if the attackers exploit zero-day vulnerabilities in some of the attacking stages."

In fact, for web traffic, the article [X] presents experimental work demonstrating that only a small percentage of web attacks (around 10%) are caught by Snort with default rules. While it is true that this percentage rises sharply to 80% when additional rules are employed, it is also true that the number of false positives generated reduces accuracy from values close to 100% with default rules to 1.5%.

Therefore, the authors are requested to include in their work this caveat, suggesting for future research that it would be beneficial to experiment with different rule sets and consider the false positives they generate in legitimate traffic.

7. PLOS authors have the option to publish the peer review history of their article (what does this mean?). If published, this will include your full peer review and any attached files.

Reviewer #4: No

---

## [Author Response · Author response to Decision Letter 2]

1 Mar 2024

Reviewer#4, Concern # 1: In fact, for web traffic, the article [X] presents experimental work demonstrating that only a small percentage of web attacks (around 10%) are caught by Snort with default rules. While it is true that this percentage rises sharply to 80% when additional rules are employed, it is also true that the number of false positives generated reduces accuracy from values close to 100% with default rules to 1.5%. Therefore, the authors are requested to include in their work this caveat, suggesting for future research that it would be beneficial to experiment with different rule sets and consider the false positives they generate in legitimate traffic.

Author response: Thank you for your valuable comments while reviewing my paper. We have incorporated the following content into the future work section of our paper as per your suggestion. The detailed changes are as follows:

…

Furthermore, the selection of alert information generated by Snort has been a consistent methodology. However, it should be noted that Snort does not have the ability to detect all attack behaviors and generate alert information. The detection capability of Snort is highly dependent on the rules used. Therefore, in our future research, we will attempt to use different sets of Snort rules and consider their false positives in legitimate traffic.

In the research on multi-stage attack detection, most researchers have chosen Snort default rules to generate alert information for different datasets[8][13][17], making their methods reproducible and easy to compare. In fact, for network traffic, Snort's default rules can only capture a small portion of attacks (10\\%), and when additional rules are used, the proportion of captured attacks increases to 80\\%, but this also reduces the accuracy from nearly 100\\% with default rules to 1.5\\%. Hence, another task for our future endeavors is to develop more insightful Snort rules, aiming to enhance data quality and, consequently, improve detection accuracy.

…

---

## [Editor Report · Decision Letter 3]

6 Mar 2024

Anomaly based multi-stage attack detection method

PONE-D-23-29167R3

Dear Dr. Hou,

We’re pleased to inform you that your manuscript has been judged scientifically suitable for publication and will be formally accepted for publication once it meets all outstanding technical requirements.

Kind regards,

Praveen Kumar Donta, Ph.D.

Academic Editor

PLOS ONE
---

## [Editor Report · Acceptance letter]

11 Mar 2024

PONE-D-23-29167R3 

PLOS ONE

Dear Dr. Hou, 

I'm pleased to inform you that your manuscript has been deemed suitable for publication in PLOS ONE. Congratulations! Your manuscript is now being handed over to our production team.

Kind regards, 

on behalf of

Dr. Praveen Kumar Donta 

Academic Editor

PLOS ONE